# Cultivation of marine bacteria of the SAR202 clade

Yeonjung Lim[1,2], Ji-Hui Seo[1], Stephen J. Giovannoni [3], Ilnam Kang [2] ✉ & Jang-Cheon Cho [1] ✉

Bacteria of the SAR202 clade, within the phylum *Chloroflexota*, are ubiquitously distributed in the ocean but have not yet been cultivated in the lab. It has been proposed that ancient expansions of catabolic enzyme paralogs broadened the spectrum of organic compounds that SAR202 bacteria could oxidize, leading to transformations of the Earth's carbon cycle. Here, we report the successful cultivation of SAR202 bacteria from surface seawater using dilution-to-extinction culturing. The growth of these strains is very slow (0.18–0.24 day$^{-1}$) and is inhibited by exposure to light. The genomes, of ca. 3.08 Mbp, encode archaella (archaeal motility structures) and multiple sets of enzyme paralogs, including 80 genes coding for enolase superfamily enzymes and 44 genes encoding NAD(P)-dependent dehydrogenases. We propose that these enzyme paralogs participate in multiple parallel pathways for non-phosphorylative catabolism of sugars and sugar acids. Indeed, we demonstrate that SAR202 strains can utilize several substrates that are metabolized through the predicted pathways, such as sugars L-fucose and L-rhamnose, as well as their lactone and acid forms.

The SAR202 clade in the phylum *Chloroflexota* is ubiquitously distributed in the ocean, accounting for 10–30% of planktonic prokaryotes in the deep sea[1–7]. Various properties associated with organoheterotrophy and sulfur and nitrogen metabolism have been interpreted from SAR202 metagenome assemblies and single-cell genome sequences[6,8–12]. The seven groups (subclades) of SAR202 are individually distinct in the numbers and types of paralogs they contain[8,12], suggesting a relationship between paralog evolution and niche specialization of the subclades.

Paralogous flavin-dependent monooxygenase genes in group III SAR202, in some cases exceeding 100 per genome, are proposed to have evolved to harvest carbon and energy from diverse organic molecules that accumulated in the oceans during the expansion of the Earth's carbon cycle, following the rise of oxygenic phototrophy[8,12,13]. Similarly, in group I SAR202, large expansions of paralogs in the enolase protein superfamily are proposed to have evolved to enable these cells to metabolize compounds that resist biological oxidation because of their chiral complexity. The early branching of these paralogs in phylogenetic trees indicates that SAR202 cell evolution was a crucible for their diversification[8,12,13].

SAR202 cells are found throughout ocean water columns, reaching highest numbers near the ocean surface, but they contribute a higher percentage of all plankton cells in the meso-, bathy-, hadal-, and abyssopelagic[5,10,12]. Group I and II SAR202 have rhodopsin genes in their genomes and are the most abundant SAR202 in epipelagic environments, whereas group III, lacking rhodopsin genes, is largely responsible for the high relative abundance of SAR202 in the dark ocean.

The cultivation of unrepresented cell types is a priority for microbiologists because cells frequently exhibit properties that cannot be easily predicted from their genomes[14]. A recent study has shown that genome-based inference fails to reliably predict catabolic pathways for more than 50% of carbon sources utilized by diverse prokaryotes that have well-curated phenotypic data[15]. Despite the recent

[1]Department of Biological Sciences and Bioengineering, Inha University, Incheon 22212, Republic of Korea. [2]Center for Molecular and Cell Biology, Inha University, Incheon 22212, Republic of Korea. [3]Department of Microbiology, Oregon State University, Corvallis, OR 97331, USA. ✉e-mail: ikang@inha.ac.kr; chojc@inha.ac.kr

surge in interest and technological advances in cultivation, a high proportion of prokaryotic groups remain uncultured[16]. The SAR202 clade has no cultured isolates yet, making it one of the "most wanted" in culture and a key "target for cultivation"[14,17].

Here we report the successful cultivation of SAR202 bacteria. Twenty-four isolates of subclade I were retrieved from surface seawater samples by dilution-to-extinction in sterile seawater media. Metabolic reconstruction supported by experimental data with cells implicated enolase and dehydrogenase paralogs in non-phosphorylative sugar oxidation. We propose that multiple parallel metabolic pathways of this type enable these cells to harvest complex mixtures of sugar-related compounds from dissolved organic carbon pools. SAR202, which are found throughout the water column of modern oceans, evolved concurrently with the rise of oxygenic phototrophy[13]. We propose they expanded into the niche of harvesting dilute and diverse carbohydrate-related molecules as the oceans are oxidized.

## Results and discussion

### The successful cultivation of the SAR202 clade

Dilution-to-extinction experiments with low-nutrient heterotrophic media (LNHM; Supplementary Table 1) in microtiter dishes retrieved twenty-four SAR202 group I isolates from surface samples (depth, 10 m) from two stations (GR1 and GR3) located nearby Garorim Bay of the Yellow Sea (Supplementary Fig. 1a). Four conditions, differing by catalase addition and light exposure (continuous dark vs. 14:10 h light-dark cycle) yielded 610 strains, of which 24 belonged to the SAR202 clade (Supplementary Table 2). All 24 SAR202 strains were obtained from cultures incubated continuously in dark (Supplementary Table 2).

Phylogenetic comparisons showed that the isolates had nearly identical 16S rRNA gene sequences, were affiliated with the SAR202 group I (Fig. 1a)[3,8,12], and corresponded to major (greater than 68%) amplicon sequence variant (ASV) of the SAR202 clade (0.7–1.0% in total prokaryotes) in the water samples (Supplementary Fig. 2).

### The growth of SAR202 isolates was very slow and inhibited by light

Four strains selected for further experiments, JH545, JH702, JH639, and JH1073, behaved similarly, growing slowly (0.18–0.24 day$^{-1}$) and displaying sensitivity to light (Fig. 2a–c). Strain JH545 grew optimally at 15–20 °C (Supplementary Fig. 3), and therefore all subsequent experiments were performed at 20 °C. Approximately 50 days were required to reach stationary phase at maximum cell densities of ~2 × 10$^8$ cells mL$^{-1}$ (Fig. 2a). Very little growth followed by gradual decline at 4 °C (Supplementary Fig. 3) suggests that strain JH545 belonging to the SAR202 group I may not be well-adapted to deep sea, where other members of the SAR202 clade (e.g., group III) are prevalent[12].

The growth of all four strains was inhibited by light-dark cycles with broad spectrum LED lights (Fig. 2b), and experiments with strain JH545 showed that continuous exposure to light caused growth inhibition followed by cell death at all light intensities tested (~45–134 µmol photons m$^{-2}$ s$^{-1}$), with variations in response levels (Fig. 2c).

Although the isolates were pure cultures, TEM and SEM microscopy showed short rods (~0.8 × 0.4 µm), cocci (diameter, ~0.5 µm), discs, and discs with biconcave centers that sometimes appeared to be toroidal, as reported for the strains of *Dehalococcoides*[18,19], a genus belonging to the same class (*Dehalococcoidia*) as SAR202 (Fig. 2d–f and Supplementary Fig. 4). Thin-section TEM images indicated monoderm cell envelopes, similar to other *Chloroflexota* (Fig. 2e)[20,21].

### Genomic features

**General genome features and phylogenomics.** The genomes of the four SAR202 strains were 3083–3094 kb in length with 51.8% GC content, ~87.5% coding density, and at least 99.9% average nucleotide identity (ANI) among the strains. The two genomes sequenced on PacBio platform (JH545 and JH1073) were assembled into one circularly closed contig, whereas the other two genomes sequenced with Illumina technology were composed of more than 30 contigs (~846 kb of N50 for both genomes) (Supplementary Fig. 5c). The circular map of the JH545 genome showed several regions with anomalous GC content and GC skew (Supplementary Fig. 5a), many of which overlapped with the predicted genomic islands (Supplementary Fig. 5b).

Genome-inferred metabolic features were nearly identical among the four genomes (e.g., COG profiles; Supplementary Table 3), leading us to focus on one of them, JH545, for further analysis (Fig. 3). In accord with previous studies, the genome annotation indicated central carbon and energy metabolism typical of aerobic organoheterotrophs, with some genes indicating capacities for lithotrophy by sulfide oxidation (sulfide:quinone oxidoreductase) and anaerobic respiration by nitrate reduction (NapAB) and N$_2$O reduction (NosZ) (Supplementary Notes). The presence of NapAB and NosZ has been reported in the SAR202 MAGs obtained from the northern Gulf of Mexico "dead zone", where the expression of these genes was detected in the samples with lowest dissolved oxygen[9]. In agreement with previous genome analyses of *Dehalococcoidia* members[19,22,23], peptidoglycan biosynthesis was not encoded in the SAR202 genomes. The TEM images showed a layer outside of the cell membrane of JH545 (Fig. 2d, e), reminiscent of the S-layer observed in peptidoglycan-lacking *Dehalococcoidia* strains[24,25].

Whole genome phylogenies of the Genome Taxonomy Database (GTDB) indicate that the SAR202 group I cells we cultured represent isolates of a hitherto-uncultured order (UBA1151) within a monophyletic superorder comprised of all SAR202 (Fig. 1b and Supplementary Notes). We propose the provisional taxonomic name "*Candidatus* Lucifugimonas marina", which includes strains JH639, JH702, and JH1073, in addition to strain JH545 as the type strain. To accommodate this novel genus and species, we also propose the family "*Candidatus* Lucifugimonadaceae" fam. nov. and the new order "*Candidatus* Lucifugimonadales" ord. nov. within the class *Dehalococcoidia* (see Methods section).

**Archaellum.** A gene cluster for archaellum, an archaeal motility structure, was predicted in the four SAR202 genomes. This gene cluster is similar to conserved arrangements of archaellum genes observed in archaea[26,27] and includes six tandem copies of *flaB* (encoding archaellin), followed by the genes *flaGFHIJ* (Supplementary Fig. 6). Six additional copies of *flaB* genes were scattered throughout the genome. A candidate gene for FlaK, a family of prepilin peptidases that remove signal peptides from archaellin[26], was also predicted based on the assignment to COG1989 and the presence of domain PF01478 (Peptidase_A24; Type IV leader peptidase family)[28].

Archaella are related to type IV pili and have no evolutionary relationship to bacterial flagella. Although a *Chloroflexota* MAG from aquifer sediment was previously reported to harbor a gene cluster for archaellum[29], bacterial isolates have never been reported to harbor archaella. Searches for FlaB homologs in the IMG database revealed that two SAR202 MAGs from the Gulf of Mexico had archaellum gene clusters very similar to that of JH545 (Supplementary Fig. 6)[9]. Exploration of FlaB (K07325) distribution using the AnnoTree database showed that most (45 of 52) bacterial genomes harboring *flaB* were affiliated to *Chloroflexota*. The remaining seven bacterial genomes having *flaB* were distributed among seven phyla, indicating that this gene is very rare among other bacteria. Of the 45 *flaB*-harboring *Chloroflexota* genomes, 40 were among the 212 *Dehalococcoidia* genomes in the database, and the remaining 5 were in *Anaerolineae*. This distribution suggests that archaella genes might have been transferred from *Archaea* to an ancestor of *Chloroflexota* and retained in the class *Dehalococcoidia*. A phylogenetic analysis of FlaB sequences found in strain JH545, several *Chloroflexota* genomes, and representative archaeal isolates, confirmed this inference. The FlaB sequences of

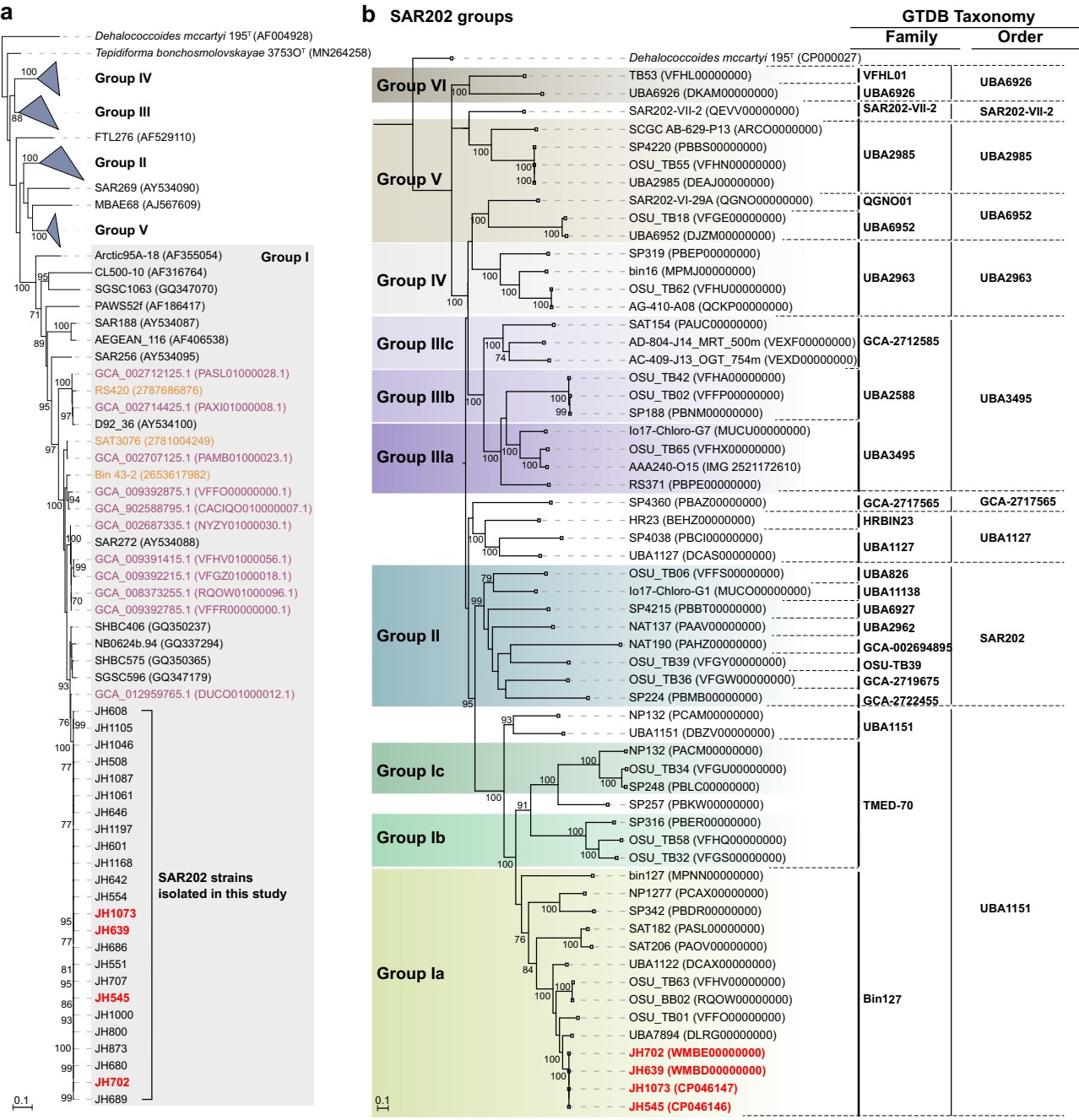

**Fig. 1 | Phylogenetic position of SAR202 strains isolated in this study.**
**a** Phylogenetic tree based on 16S rRNA gene sequences showing the relationship among the 24 SAR202 strains isolated in this study and closely related sequences retrieved from various databases, including GTDB (marked in purple), IMG (yellow), GenBank, and EzBioCloud. The four SAR202 strains selected for genome sequencing are marked in red. RAxML (v8.2.12) was used for tree building with GTRGAMMA model. Bootstrap supporting values (≥70%; from 100 resamplings) are indicated. *Dehalococcoides mccarty* was set as an outgroup. Bar, 0.1 substitutions per nucleotide position. **b** Phylogenomic tree of the SAR202 clade. The four genomes of SAR202 strains from this study (marked in red) and other MAGs and SAGs from previous studies were classified using GTDB-Tk. Species cluster representative genomes of the GTDB taxa, into which the analyzed SAR202 genomes were classified, were included for tree building. Designation of the SAR202 subgroups following the classification scheme of a previous study[12] is indicated on the left side of the tree with color shadings. On the right side of the tree, taxonomic assignment of the genomes at the family and order levels according to the GTDB (release 202) is shown. The tree building was performed using RAxML (v8.2.12), with PROTGAMMAAUTO option, based on a concatenated alignment of core genes obtained by UBCG pipeline. Bootstrap supporting values (100 iterations) are indicated on the nodes. Bar, 0.1 substitution per amino acid position.

*Chloroflexota* formed a monophyletic clade, which was located as a sister clade of an archaeal FlaB group (Supplementary Fig. 7). Investigation on the 45 *Chloroflexota* genomes having FlaB (found in Anno-Tree) and BlastP searches at NCBI (FlaB of strain JH545 as a query) revealed that at least two *Chloroflexota* strains harboring archaella gene cluster (flaB and several other genes) have been cultivated:

*Litorilinea aerophila*[30,31] and *Aggregatilinea lenta*[32] (Supplementary Fig. 6). No archaella were observed during microscopic examination of the SAR202 cells.

**Heliorhodopsin.** Two copies of heliorhodopsin (HeR) genes were detected in the JH545 genome. In a recent study of marine SAR202

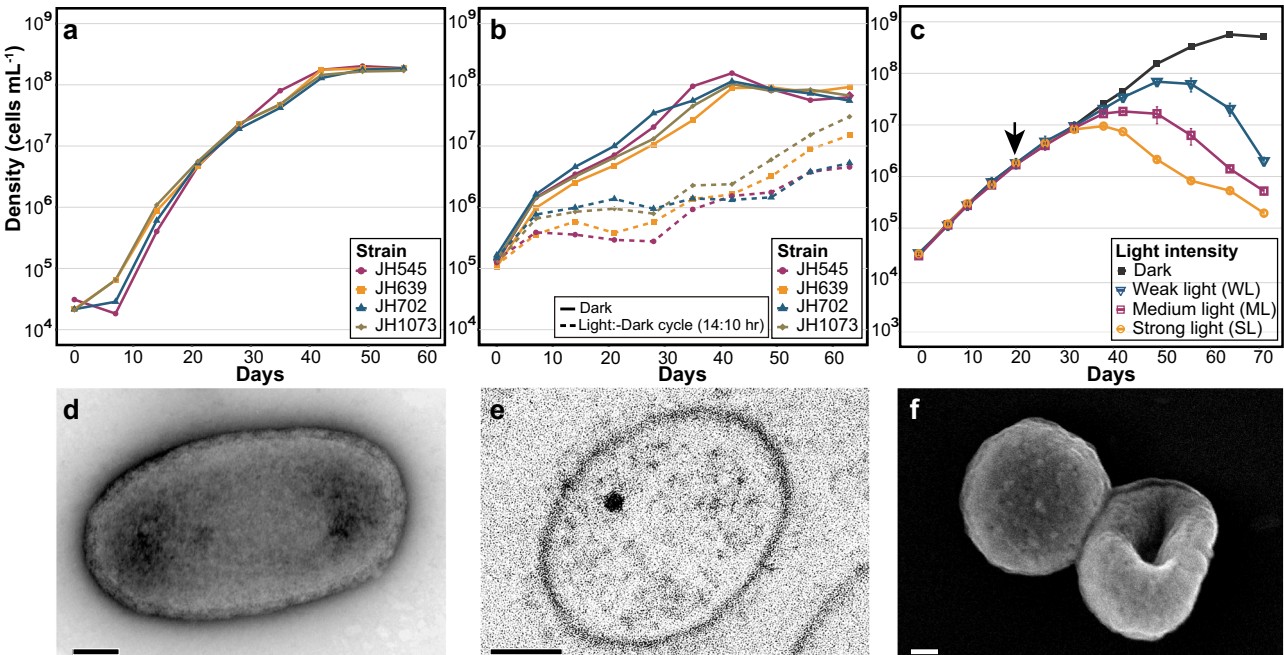

**Fig. 2 | Growth and morphology of SAR202 isolates. a** Initial growth curves of four representative SAR202 strains in low-nutrient heterotrophic medium (LNHM) under dark condition. **b** Growth curves of the four strains under continuous dark and light-dark conditions in LNHM. Light-dark cycle (14:10 h) was applied using LED lamps with a light intensity of ~155 μmol photons m$^{-2}$ s$^{-1}$. For the dark condition, tubes were wrapped in aluminum foil. **c** Growth curves of strain JH545 under dark condition and various light intensities in LNHM5x. All cultures were initially incubated in dark (i.e., the tubes were wrapped in aluminum foil). At the day indicated by a black arrow, some cultures were shifted to light condition (i.e., the aluminum foil was removed) and exposed to light with the following intensities: WL, ~45; ML, ~89; SL, ~134 μmol photons m$^{-2}$ s$^{-1}$. The experiment in **c** was performed in triplicates. Data are presented as mean values ± SEM. Note that the error bars are hidden when they are shorter than the size of the symbols. **a**–**c** Source data are provided as a Source data file. **d**–**f** Electron micrographs of strain JH545 cells observed by TEM (**d**), thin-section TEM (**e**), and SEM (**f**). Scale bars, 100 nm.

MAG/SAGs, rhodopsin genes were found in 28 group I and II genomes, all retrieved from water depths of less than 150 m. An HeR gene was reported in a single group II genome[12]. Therefore, this finding represents the presence of HeR in SAR202 group I. The two HeR copies found in the JH545 genome exhibited ~83% amino acid identity and were located one gene downstream of DNA photolyase, which repairs DNA damage caused by UV exposure using visible light (Supplementary Fig. 8a). Multiple sequence alignment showed that the two copies differed in the amino acid residue that caused a spectral shift in an Ala scanning mutagenesis study[33] (W163 in 48C12; Supplementary Fig. 8b), suggesting that the two HeR copies might have different absorption spectra. Although the function of HeR remains unresolved, it has recently been suggested that it might function as a light sensor that regulates responses to light-induced oxidative stress[34,35]. Given that no HeR has been found in SAR202 group III, which is abundant in the dark ocean[12], the possession of HeR by JH545 isolated from coastal surface water may indicate adaptation to the euphotic habitat.

**Transporters and sulfatases.** In accord with a previous study[9], a large number of major facilitator superfamily (MFS) transporters were found in the JH545 genome, including 42 proteins assigned to COG0477 (Supplementary Table 3). MFS is a very large family of membrane transporters that are known to transport a variety of compounds, including mono- and oligosaccharides, amino acids, and nucleosides[36,37]. It is noteworthy that some substrates that we report below enhance the growth of JH545 (see the next section on COG4948) are known to be transported by MFS proteins[38–40].

Eighteen proteins in the JH545 genome were assigned to COG3119, annotated as arylsulfatase A or a related enzyme (Supplementary Table 3). Sulfatase paralogs have been reported previously in SAR202 groups I and II[12]. Arylsulfatases catalyze the desulfation of sulfated

carbohydrates in some catabolic pathways, for example the degradation pathway of ulvan by a marine bacteria[41].

### Genomes of SAR202 group I have the highest proportion of COG4948 paralogs among all prokaryotes, and these paralogs are highly divergent

The genomes of cultivated SAR202 we report encoded 80 COG4948 proteins in the mandelate racemase family within the enolase superfamily (~2.8% of CDS; Supplementary Table 3). Many marine bacteria have minimal genomes with few paralogs, so the unusually large sets of paralogs present in diverse SAR202 groups, such as COG4948 and COG2141 in the groups I and III, respectively, attracted attention when they were first discovered[12].

We sought to establish whether the expansion of COG4948 in SAR202 group I is unusual in the context of prokaryotic diversity. We analyzed the genomes in GTDB, one of the most phylogenetically comprehensive genome databases. Because COG annotation is not scalable, we counted the numbers of the two Pfam domains (PF02746 and PF13378) corresponding to COG4948 (see Materials and Methods for details) and calculated the proportion of COG4948 proteins among all CDSs in each genome. Among the 47,894 species cluster-representative genomes of GTDB (R202), all 48 SAR202 group I (o_UBA1151 in GTDB) genomes ranked in the top 66 except one that ranked the 134th, in proportions of COG4948 (Fig. 4a), demonstrating that the paralog expansion of COG4948 is a prominent feature of SAR202 group I across all prokaryotes.

We analyzed sequence divergence among the 80 COG4948 proteins of the JH545 genome by building a sequence similarity network (SSN) using the Enzyme Function Initiative's Enzyme Similarity Tool (EFI-EST)[42]. Most of the ~70 clusters that contained JH545 proteins as their members included only one JH545 gene, indicating that the JH545 COG4948 proteins are highly divergent (Fig. 4b). While many of the

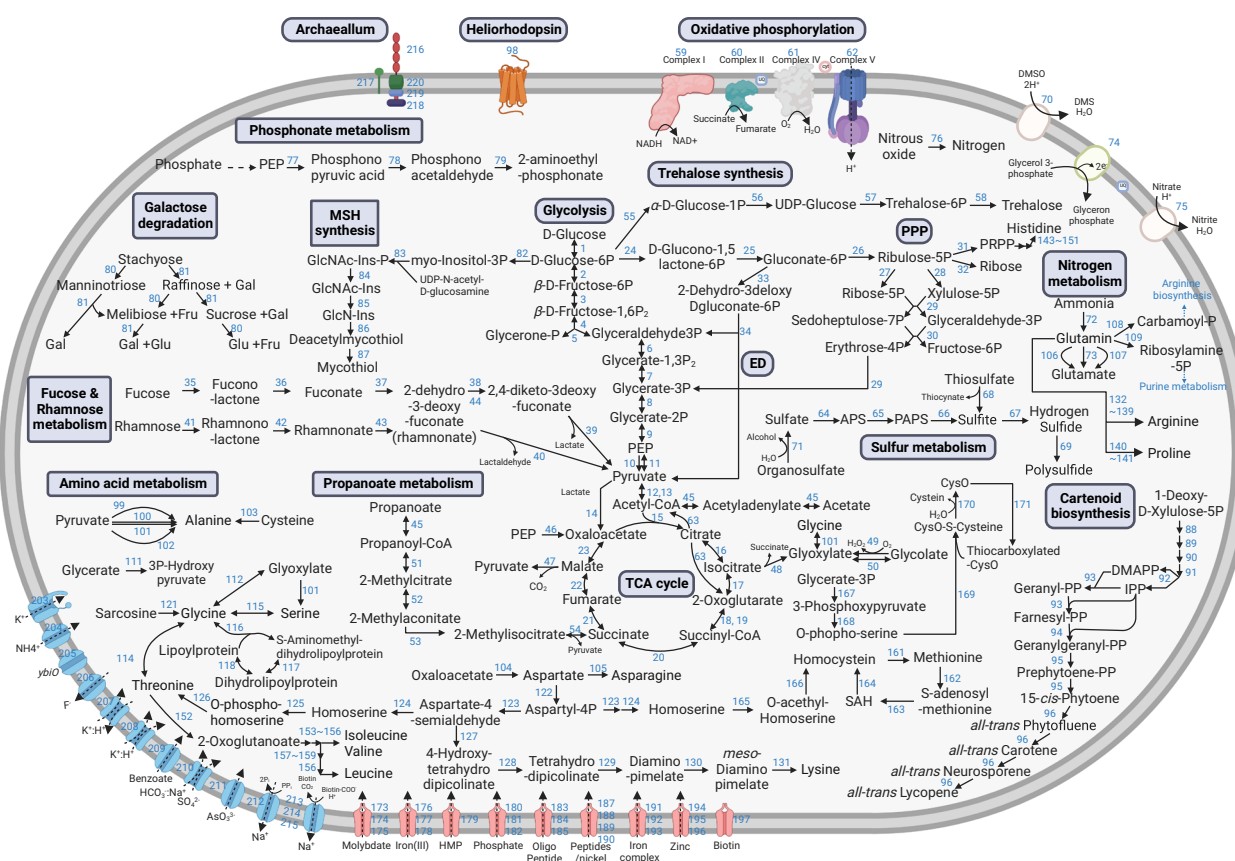

**Fig. 3 | Metabolic pathways of strain JH545.** Some key metabolic pathways are indicated by blue boxes. Detailed information on the enzymes corresponding to the numbers next to arrows is available in Supplementary Data 1. The colors of the transporters at the bottom indicate functional categories: pink, ABC transporters; blue, others. Figure created with BioRender.com.

largest COG4948 clusters included diverse bacterial phyla, many small clusters were comprised mainly of *Chloroflexota*, suggesting that the COG4948 protein family, which is distributed widely across prokaryotes, diversified in *Chloroflexota*. Only the largest cluster, which included two JH545 proteins, contained biochemically studied proteins[43] (Fig. 4b).

### Abundant COG paralogs of SAR202 participate in the degradation of sugars, lactones, and sugar acids

We propose that seven sets of paralogs (COG1028, COG0667, COG3618, COG4948, COG1063, COG3836, and COG0329), including the most abundant COG4948 paralogs, act concertedly in parallel pathways that harvest energy from diverse carbohydrates (Fig. 5b), and we provide experimental evidence that cultured SAR202 group I cells respond to predicted substrates of these pathways (Fig. 5a). The largest COG sets in the SAR202 genomes of this study, in addition to the mandelate racemases (COG4948), included NAD(P)-dependent dehydrogenases (COG1028), aldolases (COG3836), and oxidoreductases (COG0667; Supplementary Table 3). These enzymes are found in non-phosphorylative pathways of various sugars and their acid catabolism (e.g., fucose, rhamnose, arabinose, and xylose)[39, 44–49]. Central to these pathways is the pairing enzymes from the two largest paralog sets, mandelate racemase-like enzymes and NAD(P)-dependent dehydrogenases, which catalyze a dehydration reaction followed by an oxidation reaction resulting in a flow of electrons for respiration. A previous SAR202 pangenome analysis of variation in COG copy number found significant positive correlations in the numbers of copies of the same seven COGs (Supplementary Table 3) in SAR202 group I genomes relative to other SAR202 genomes[12]. The

phylogenetically correlated distributions of these paralog expansions support our prediction that they play coordinated metabolic roles.

We reconstructed non-phosphorylative pathways for the oxidation of two sugars relevant to marine environments, fucose and rhamnose[50, 51], in the genomes of the cultured SAR202 strains based on several previous studies[46,47,52–55] (Fig. 5b). Although both rhamnose and fucose are known to be degraded via pathways involving phosphorylation in many organisms, these kinase-dependent pathways were not found in the genomes. This suggests that the reconstructed non-phosphorylative pathways shown in Fig. 5b might serve as the only catabolic routes for both sugars in these strains. In the pathway reconstruction, L-fucose and L-rhamnose share the same COG annotation at each step, yielding pyruvate and lactate or lactaldehyde as final products. Seven of the eight COGs represented in the reconstructed pathways for fucose and rhamnose catabolism were from the abundant paralog sets described above, ranging from 9 to 80 variants of each COG (Supplementary Table 3).

We tested the growth response of JH545 cells to external metabolites that were predicted to be substrates for the catabolic pathways proposed above. In these experiments we used a defined medium based on artificial seawater, to which we added the same cofactors and organic compounds that were used for the original isolation of the strains. We did not examine whether the tested substrates could serve as sole carbon sources because of the very low growth rate of the cells and because cells adapted to oligotrophic ecosystems often exhibit reduced metabolic flexibility in comparison to copiotrophic cells when challenged with simplified carbon mixtures. All substrates tested, including L-fucose, L-rhamnose, their lactone and acid forms, and ascorbate

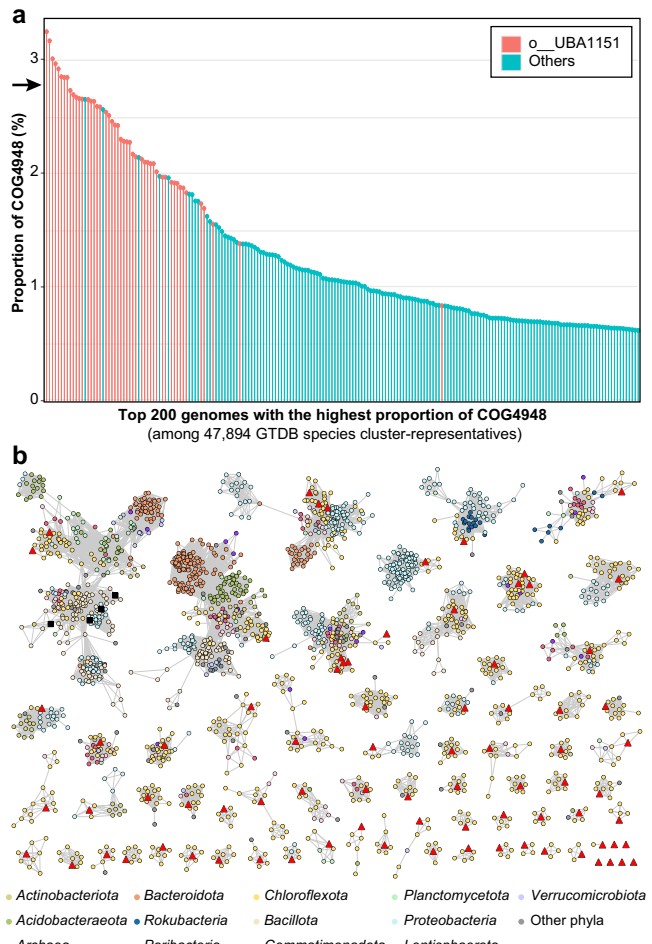

**Fig. 4 | High proportion and diversity of COG4948 proteins in the SAR202 clade. a** Proportion of COG4948 proteins among the species cluster-representative genomes of GTDB (release 202; 47,894 genomes). Only the top 200 genomes with the highest proportion are included. Genomes of o_UBA1151 (SAR202 group I) are indicated with a distinct color. All 48 o_UBA1151 genomes are included in this figure owing to the high proportion of COG4948. The black arrow on the y-axis indicates the proportion of the four SAR202 genomes sequenced in this study (~2.8%). Source data are provided as a Source data file. **b** SSN (sequence similarity network) analysis of diverse COG4948 proteins predicted in the JH545 genome. UniProt proteins with the two Pfam domains, PF02746 and PF13378, were included in the analysis together with the 80 COG4948 proteins of strain JH545. Only the protein clusters including the JH454 proteins were retained in this figure after applying a threshold cutoff value to the alignment score (≥150). Nodes (proteins) are colored by their phylum-level taxonomy, following the color codes at the bottom. The 80 COG4948 proteins of the JH545 genome are indicated in red triangles. The four proteins that have been studied experimentally are indicated with black rectangles (UniProt ID: D8ADB5, C9A1P5, C6CBG9, and A8RQK7).

enhanced the growth of strain JH545 in artificial seawater media (Fig. 5a). With increases in growth rates, cell densities in late exponential phase (~35 days of incubation) were more than 10 times higher in substrate-amended cultures, although the cultures reached similar densities in stationary phase. Ascorbate was tested because an ascorbate degradation pathway requiring a COG4948 enzyme[39,48] was nearly complete in the JH545 genome (Supplementary Fig. 9).

## SAR202 isolates represent a cell type that is common in the euphotic zone but relatively more abundant in the dark ocean

The vertical distribution of species-level population represented by JH545 (hereafter, JH545 population) in marine metagenomes followed patterns previously observed for several SAR202 group I members[12]. In metagenomes from *Tara* Oceans and station ALOHA, the relative abundance of the JH545 population was higher in the mesopelagic zone (200–1000 m) compared to the euphotic zone (surface to 200 m) (Mann–Whitney *U* test, $P = 1.78 \times 10^{-10}$, one-sided; Fig. 6c). In metagenomes from several marine trenches, the relative abundance of the JH545 population generally increased with increasing depth above the hadopelagic zone (below 6000 m), where their relative abundance declined (Fig. 6a, b). Given the overall decline of cell numbers with increasing water depth[56–58], the JH545 population is likely found throughout the ocean water column, reaching its highest concentration in the epipelagic zone but increasing in relative abundance in the dark ocean. This inference is also consistent with recent studies reporting higher SAR202 cell abundance in the euphotic zone compared with the aphotic zone in the Atlantic Ocean and Fram Strait as measured by FISH-based cell counting[12,59]. The likely high absolute abundances of the JH545 population observed in the euphotic zones seemingly conflict with the observations of growth inhibition by light-dark cycles and death in continuous light of the cultured strains upon exposure to broad spectrum white light (Fig. 2b, c). To reconcile these observations, we hypothesize that JH545 uses its two copies of heliorhodopsin to regulate functions that are negatively impacted by light, mitigating its susceptibility to inhibition[34], and we note that the action spectrum for light inhibition has not been determined and the cells could be sensitive to frequencies that are normally absorbed in the water column. Regardless, our findings indicate that the challenge of culturing these cells might in part be explained by their sensitivity to light.

We propose that the vertical distribution of SAR202 group I members represented by the JH545 population can be reconciled with their metabolic features that we report here. The compounds that enhanced the growth of JH545 (e.g., fucose and rhamnose) and related compounds that we predict JH545 and SAR202 group I bacteria might also utilize are found in the surface ocean largely as monomers in polysaccharides produced by phytoplankton[51]. The JH545 genome had a limited repertoire of glycoside hydrolases (GHs) and lacked representative GH families annotated as fucosidase and rhamnosidase (e.g., GH29, GH95, GH141, GH78, and GH106)[41, 50,60]. SAR202 group I may scavenge monomeric compounds and their metabolic products that diffuse into the water column during degradation of polysaccharides by other taxa (e.g., *Bacteroidia*, *Gammaproteobacteria*, and *Verrucomicrobiota*) (ref. 51 and references therein), many of which are capable of much more rapid growth and would have a competitive advantage as specialists when polysaccharides are available. In the dark ocean, where polysaccharides are depleted, however, paralogous gene expansions may provide group I with a competitive advantage by allowing them to use a diverse range of sugars, sugar acids, and related compounds, leading to the observed increase in relative abundance. Piezotolerance of SAR202 cells would also give them a competitive advantage in the dark ocean[61]. We note, however, that there are other SAR202 groups (e.g., group III) that are known to contribute substantially to the high relative abundance of the SAR202 clade in the deepest ocean regions. The isolation and characterization of a wider diversity of SAR202, especially lineages typical of the dark ocean, could propel future research that aims to reconstruct carbon chemistry and ecology in the dark ocean.

Evolutionary diversifications of paralogous enzymes in SAR202 cells have been proposed to benefit these cells by expanding the range of organic compounds they can metabolize[8,12]. To explain the non-phosphorylative pathways for sugar oxidation we report, and the possibility of a much larger set of parallel pathways in the same cells, we hypothesize that SAR202 group I cells have evolved to exploit relatively rare carbohydrate compounds that are not harvested by taxa specializing in more common carbohydrate types. Carbohydrates are structurally complex because of the chirality of monomers, the

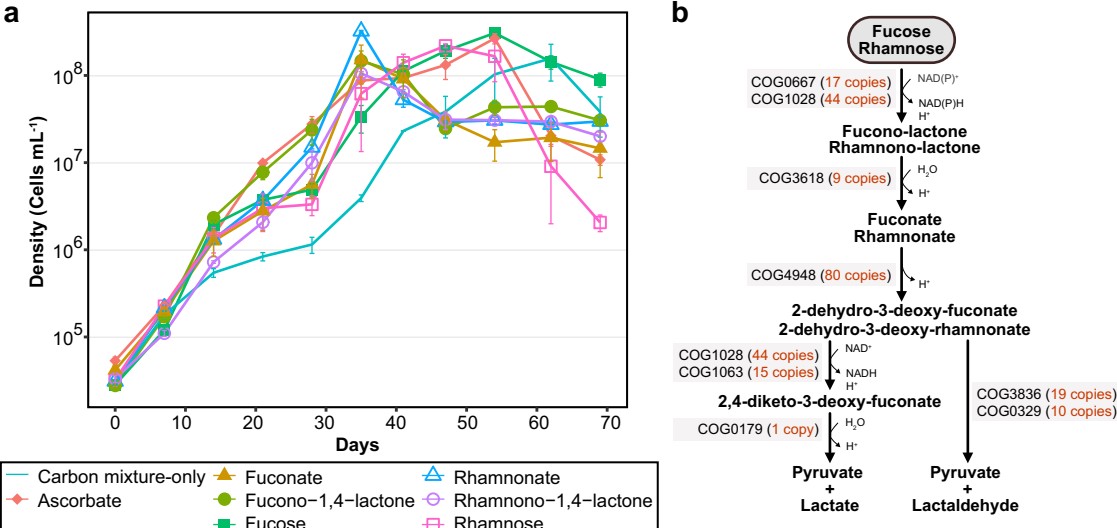

**Fig. 5 | Growth curves of strain JH545 with various carbon compounds and putative non-phosphorylative catabolic pathways of L-fucose and L-rhamnose.** **a** Growth of strain JH545 on various carbon compounds that were predicted to require COG4948 enzymes for metabolization. Carbon compounds were added to the ASW5x media (Supplementary Table 1) at a final concentration of 250 μM. Carbon mixture-only, no additional carbon compounds were added, except for carbon mixture already included in the ASW5x media. The incubation was performed in dark at 20 °C. All experiments were performed in triplicates. Data are presented as mean values ± SEM. Note that the error bars are hidden when they are shorter than the size of the symbols. Source data are provided as a Source data file. **b** Putative non-phosphorylative catabolic pathways of L-fucose and L-rhamnose inferred from the SAR202 genome data. The number of proteins assigned to the COGs is indicated within parentheses next to the COG IDs at each step.

diversity of the linkages they form, and modifications such as *O*-methylation and sulfation. In this scenario, SAR202 exploits a niche, harvesting a class of compounds that are recalcitrant because of their low abundance and structural diversity. This combination of features, i.e., high molecular complexity and low concentrations of individual molecular species, is associated with the molecular diversity hypothesis[62], a leading explanation for the sequestration of ocean carbon in dissolved molecules with millennial turnover times. Our findings are not contrary to this hypothesis; they instead suggest a class of compounds that would accumulate for the same reasons if a particular cell type, such as SAR202 group I, had not evolved a unique mechanism to harvest them. It should be noted that carbon pools in the dark ocean could also be affected by other SAR202 groups. For example, SAR202 group III, which is known to be more abundant than group I in the dark ocean, has been suggested to contribute to the degradation and transformation of recalcitrant organic matter using an expanded repertoire of flavin-dependent monooxygenases[8,12].

In summary, we describe the accomplished cultivation of the abundant and ubiquitous marine bacterial SAR202 clade, a monophyletic superorder within the class *Dehalococcoidia* of the phylum *Chloroflexota*. The SAR202 strains grew very slowly, and their growth was further inhibited by exposure to light; these properties might explain why they have not been cultivated previously. We show that these cells contain paralog expansions and that these paralogs can be arranged into non-phosphorylative pathways for the catabolism of sugars and their lactone and acid forms. We show that fucose and rhamnose, predicted substrates of these pathways, enhanced the growth of the SAR202 isolate.

We explored the physiology of a superorder of bacteria that have been implicated in the oxidation of a variety of forms of semi-labile organic carbon, and it seems likely that further studies of the diverse cells in this clade will contribute to a more mechanistic understanding of the ocean carbon cycle. Cell cultures of novel prokaryotes provide opportunities to study a wide range of cellular properties that cannot be determined from genomes alone. Interactions of these slowly growing cells with organic carbon exometabolites, and their unexplained sensitivity to light, are promising avenues for future work.

Studies of SAR202 archaella function and integration with the cell envelope of SAR202 may provide clues into the ecology and cell architecture of these unusual cells. The findings we report provided surprising insights into challenges that long-stalled SAR202 cultivation. Whether the diversity of not-yet-cultured lineages of SAR202 cells inhabiting the dark ocean can be explained by similar properties remains to be seen.

## Methods
### Sample collection and cultivation
Seawater samples used for high-throughput culturing (HTC) based on dilution-to-extinction and amplicon analysis were collected from a depth of 10 m at two stations (GR1 and GR3) in the West Sea of Korea (Yellow Sea) in October 2017 (Supplementary Fig. 1a). Physicochemical properties of the water samples are presented in Supplementary Fig. 1b. The total prokaryotic number was determined by counting 4',6-diamidino-2-phenylindole (DAPI)-stained cells using an epifluorescence microscope (Nikon 80i, Nikon) after filtration using a 0.2-μm pore-sized polycarbonate membrane filter (Millipore). Culture media for HTC (LNHM) was prepared using a seawater sample collected from the East Sea (depth, 10 m) in 2016. In brief, the seawater sample was filtered using a 0.2-μm pore-sized polyethersulfone membrane filter (Pall), autoclaved (1.5 h), sparged with $CO_2$ (8 h), aerated (24 h), and amended with carbon sources, macronutrients (nitrogen and phosphorus sources), trace metals, vitamins, and amino acids (Supplementary Table 1). The seawater samples were then diluted with culture media to a concentration of 5 cells mL$^{-1}$ and dispensed (1 mL per well) into 48-well microplates (BD Falcon). The plates were incubated at 20 °C for 1 month in dark or under LED light (Philips; correlated color temperature, 3000 K; light intensity, ~155 μmol photons m$^{-2}$ s$^{-1}$) with a light/dark cycle of 14:10 h. The microbial growth in each well was screened by flow cytometry (GUAVA EasyCyte Plus flow cytometer and Guava CytoSoft (v5.3), Millipore) after staining with SYBR Green I (Life Technologies) and recorded as growth-positive when more than $5.0 \times 10^4$ cells mL$^{-1}$ were detected. Cultures from growth-positive wells were used for further analyses and stored as glycerol stock (10%, v/v) at −80 °C.

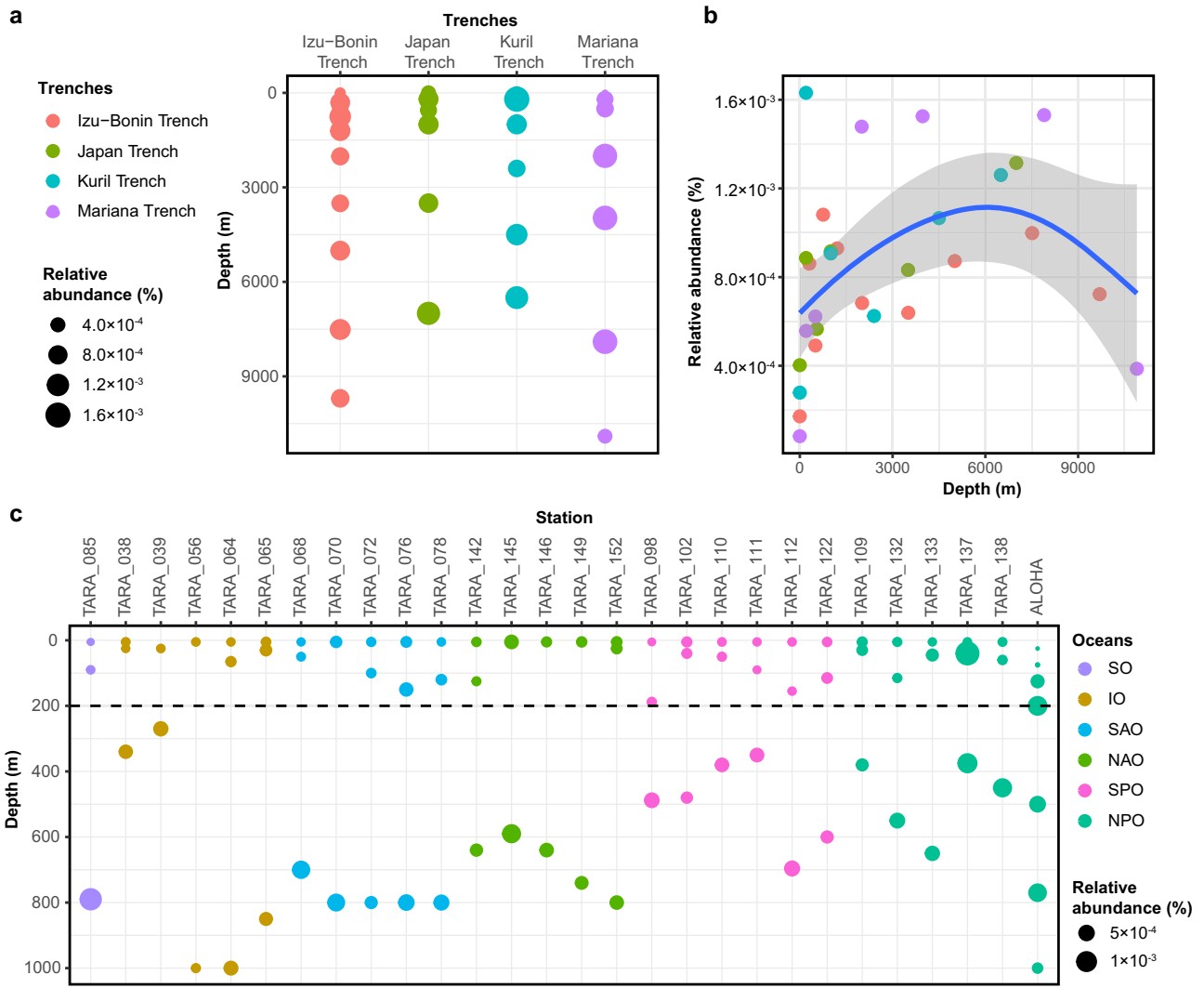

**Fig. 6 | Metagenome fragment recruitment to the JH545 genome. a, c** Relative abundances are plotted for various stations and depths from the four marine trenches with water depths of more than 6000 m (**a**) and station ALOHA and *TARA* Oceans stations (**c**). For *TARA* Oceans, only the stations with samples from water depths of both less than 100 m and more than 250 m were used. The dotted horizontal line in (**c**) indicates a depth of 200 m, the boundary between epipelagic (euphotic) and mesopelagic zones. Bubble colors in (**c**) correspond to the oceans where the sampling stations are located, following the legend at the right: SO, the Southern Ocean; IO, the Indian Ocean; SAO and NAO, the South and North Atlantic Ocean, respectively; SPO and NPO, the South and North Pacific Ocean, respectively. **b** Data used in (**a**) re-plotted to show the change in relative abundances as a function of depth. Regression line (in blue) and confidence intervals (95%; gray shading) were drawn by "geom_smooth" function of the ggplot2 R package with "gam" method (generalized additive model). Source data are provided as a Source data file.

## Phylogenetic analysis and classification of 16S rRNA gene sequences

Phylogenetic analyses of growth-positive cultures were based on PCR amplification and sequencing of 16S rRNA genes. DNA templates for PCR were prepared from growth-positive cultures using the InstaGene™ Matrix (Bio-Rad) according to the manufacturer's instructions. The amplification of 16S rRNA genes were performed by PCR using 27F and 1492R primers, followed by Sanger sequencing with 800R and 518F primers (Macrogen Inc., Korea). Taxonomic classification of 610 strains obtained from the HTC experiments was carried out using the "classify.seqs" command of the Mothur software package (v1.39.5)[63] using the SILVA database SSURef NR99 (release 132)[64] as a reference. For more refined taxonomic and phylogenetic analysis of the SAR202 isolates, the 16S rRNA gene sequences were aligned using the SINA online aligner (v1.2.11, http://www.arb-silva.de/aligner)[65], imported into ARB program[66], and inserted using the ARB parsimony into the guide tree of SILVA database SSURef NR99 (release 132)[64].

After manual curation, the aligned sequences of the isolates and their phylogenetic relatives were exported with "ssuref:bacteria" filter. Maximum-likelihood phylogenetic trees were constructed using RAxML[67] (v8.2.12) with GTRGAMMA method including 100 bootstrap replicates and visualized using the MEGA software (v7.0)[68].

## Microbial community analysis

Two liters of each seawater sample (GR1 and GR3) were filtered through a 0.2-μm pore-size polyethersulfone membrane filter (Supor, Pall). DNA was extracted directly from the membrane filters using DNeasy PowerWater Kit (Qiagen) according to the manufacturer's instructions. The V4-V5 regions of 16S rRNA genes were amplified using fusion primers that were designed based on universal primers, 518F and 926R[69]. The pooled PCR products were sequenced on Illumina MiSeq platform (300-bp, paired-end; Chunlab Inc.). The analyses of the 16S rRNA gene amplicon sequences were performed using QIIME2[70] after primer trimming with cutadapt (v2.7)[71].

## Culture experiments

The SAR202 strains were grown and maintained using LNHM5x at 20 °C in dark. Growth experiments were performed using LNHM, LNHM5x, and ASW5x. The detailed recipes of the media are presented in Supplementary Table 1. Bacterial growth was monitored by flow cytometry, and the purity and identity of cultures were regularly determined by sequencing the amplified 16S rRNA genes and by microscopic examination.

The growth of strain JH545 was monitored at temperatures ranging from 4 to 37 °C in dark. Growth substrate tests were performed using L-rhamnose, L-rhamnono-1,4-lactone, L-rhamnonate, L-fucose, L-fucono-1,4-lactone, L-fuconate, and ascorbate. All tested compounds were purchased from Sigma-Aldrich. These chemicals were added to ASW5x media at a final concentration of 250 µM. Cultures were incubated in dark at 20 °C.

For the initial examination of the effect of light exposure on SAR202 growth (Fig. 2b), cells were inoculated in LNHM at an initial cell density of ~$1.0 \times 10^4$ cells mL$^{-1}$ and incubated in a chamber equipped with LED (3000 K; Phillips), with a light/dark cycle of 14:10 h. Light intensity was ~155 µmol photons m$^{-2}$ s$^{-1}$. To test the effect of light intensity on SAR202 growth (Fig. 2c), a custom-made equipment with a LED-array (3000 K) and a light dimmer was utilized, and the light intensities were set at ~45 (weak light), ~89 (medium light), and ~134 (strong light) µmol photons m$^{-2}$ s$^{-1}$. During the test, the lights continuously remained on. All experiments included control samples that were constantly kept in dark by wrapping the tubes or flasks with aluminum foil. Light intensity was measured using a digital light meter (TES-1335; TES).

## Morphological characterization

Cell morphology was observed by transmission electron microscopy (TEM; CM200, Philips) and scanning electron microscopy (SEM; S-4300 and S-4300SE, Hitachi). To prepare samples for TEM, 20 mL of culture prefixed with 2.5% glutaraldehyde were filtered using a 0.2-µm pore-size polycarbonate membrane on which formvar/carbon-coated copper grids were placed, followed by staining of the grids with uranyl acetate (2%). For thin-section TEM, centrifuged cells from 800 mL of culture were subjected to primary fixation with Karnovsky's solution (2% paraformaldehyde, 2.5% glutaraldehyde), post-fixation with 2% $OsO_4$, en-bloc staining with 0.5% uranyl acetate, sequential dehydration with 30, 50, 70, 80, 90, and 100% ethanol, and embedding with resin (EMBed-812, Electron Microscopic Science). Finally, sectioning was performed using a diamond knife. Thin sections were stained with uranyl acetate (2%). For SEM analysis, 100 mL of culture was concentrated by centrifugation, fixed with 2.5% glutaraldehyde, post-fixed with 1% $OsO_4$, dehydrated with 30, 50, 70, 80, 90, and 100% ethanol, and chemically dried using hexamethyldisilazane (Sigma-Aldrich). Treated samples were gently mounted on a cover glass and coated with a thin layer of carbon.

## Genome sequencing, assembly, and analyses

Genomic DNA of the SAR202 strains was extracted from cell pellets obtained by centrifugation of liquid cultures (~1 L), using DNeasy Blood & Tissue Kit (Qiagen), according to manufacturer's instructions. The genomic DNA of strains JH545 and JH1073 was used for the construction of the 20-kb SMRTbell library, which was sequenced on the PacBio RS II platform (Pacific Biosciences). De novo assembly of raw sequencing reads was carried out by the RS_HGAP_Assembly.2 protocol of SMRT Analysis (v2.3.0), resulting in a single contig. The contig was circularized using Circlator (v1.5.5)[72] and polished using the RS_Resequencing.1 protocol of SMRT Analysis to obtain the final error-corrected genome sequence. Genome sequencing of strains JH702 and JH639 was performed on the Illumina HiSeq platform (2 × 150 bp). Raw reads were trimmed using BBDuk with the following options: ktrim=r k=23 mink=11 hdist=1 tpe tbo ftm=5 qtrim=rl trimq=10 minlen=100.

Assembly of Illumina sequencing data was performed using SPAdes v3.11.1 in a multi-cell mode with read error and mismatch correction[73].

The genome sequences were submitted to the IMG-ER system for annotation. Prokka (v1.12)[74] was also used for annotation. The predicted protein sequences were analyzed using BlastKOALA[75] and KofamKOALA[76] for metabolic pathway reconstruction based on KEGG Orthologs (KOs). Annotation by eggnog-mapper (v2.0.1)[77] and hmmsearch (v3.3) against protein databases such as Pfam were also performed for more accurate and detailed functional annotation. Analysis of CAZymes was performed using dbCAN2[78]. A map of metabolic pathways was created using BioRender (https://biorender.com). ANIb values between genomes were calculated using JSpeciesWS[79]. Genomic islands were predicted using IslandViewer 4[80]. Genome comparison was visualized by BLAST Ring Image Generator (BRIG)[81].

The SAR202 genome sequences of the present study and a previous study[12] were classified using GTDB-Tk (v1.7.0), which indicated that the genomes belonged to ~10 different orders according to the GTDB. Representative genomes of these orders were used to construct a phylogenomic tree of the SAR202 clade. We used UBCG pipeline[82] to obtain a concatenated alignment of core genes. A maximum-likelihood tree was constructed using RAxML (v8.2.12)[67], with a PROTGAM-MAAUTO option including 100 bootstrap iterations.

## Analyses of the HeR and FlaB

Genomic regions around the HeR genes of the JH545 genome were visualized using Easyfig[83]. Multiple amino acid alignment of HeRs from JH545 and the first-characterized HeR (48C12) was performed using ClustalW[84]. The Clustal X color scheme of the Jalview (v2.11.1.3)[85] was applied to visualize the alignment.

Various FlaB sequences were collected for phylogenetic analysis. In addition to literature searches, putative orthologs of JH545 FlaBs were searched using SHOOT[86] (https://www.shoot.bio/), resulting in the retrieval of only archaeal FlaBs. Bacterial FlaBs were searched using BlastP in IMG-ER and NCBI (nr database) with JH545 FlaBs as queries. Additionally, AnnoTree was searched using K07325 (archaeal flagellin FlaB) as a query. The collected sequences were aligned using MUSCLE, followed by tree building using RAxML (v8.2.12) with the PROTGAM-MAAUTO option. FlaBs from MAGs or SAGs affiliated with bacterial phyla other than *Chloroflexota* were removed from the analyses, as these FlaBs were the only ones found in their respective phyla, suggesting the possibility of contamination. Some archaeal FlaBs that are much longer than other sequences were also removed. Some genomes in which FlaB was found were used to visualize the gene map of the archaella gene cluster using Easyfig.

## Analysis of COG4948 proteins

A total of 80 proteins were assigned to COG4948 by the IMG-ER annotation of the JH545 genome, which was also verified by the Conserved Domain search[87] (v3.18) at NCBI. A search against Pfam-A database (by Pfam_Scan.pl; both available at https://ftp.ebi.ac.uk/pub/databases/Pfam/) showed that 74 COG4948 proteins had both PF02746 and PF13378 domains. The remaining six proteins had either PF13378 (four proteins) or PF02746 (two proteins) domains. The two Pfam domains were found only in 80 COG4948 proteins. Based on these results showing a correspondence between COG4948 and the two Pfam domains, we decided to use the two Pfam domains for approximate calculation of the proportion of COG4948 proteins in 47,894 species cluster-representative genomes of the GTDB (R202). The genomic faa files (gtdb_proteins_aa_reps_r202.tar.gz) were downloaded from the GTDB repository (https://data.gtdb.ecogenomic.org/) and searched using hmmsearch (v3.3) against the hmm files of the two Pfam domains, with "cut_tc" option. The approximate proportion of COG4948 proteins in each genome was calculated by dividing the number of hmmsearch hits by twice the number of CDS. The results were visualized using the R package 'tidyverse'.

The 80 COG4948 protein sequences of JH545 were submitted to the Enzyme Similarity Tool (EFI-EST; https://efi.igb.illinois.edu/efi-est/)[42] to generate an SSN. A total of 77,142 proteins in the UniProt database (v2020_02) that had the two Pfam domains were included in the SSN, which was constructed using BLAST with default options. The obtained SSN was explored using the Cytoscape (v3.7.2)[88]. After the application of a cutoff threshold of alignment score (larger than or equal to 150), clusters that included the 80 COG4948 proteins of the strain JH545 were retained for visualization.

Reconstruction of non-phosphorylative fucose and rhamnose degradation pathways is based on a KEGG pathway map (fructose and mannose metabolism; map00051), MetaCyc pathways (ʟ-fucose degradation II/III and ʟ-rhamnose degradation II/III), and several publications[46,47,52–55]. When necessary, proteins characterized in past reports were explored in the UniProt database or analyzed by CD-search at NCBI to ascertain their COG assignments.

### Metagenome fragment recruitment

The relative abundance of the JH545 population in marine metagenomes was estimated using CoverM (v0.6.1; https://github.com/wwood/CoverM). Metagenomes from several marine trenches, station ALOHA (collected in December, 2011), and some *Tara* Oceans stations were downloaded from SRA and quality-trimmed using BBduk (v38.86)[89]. Ribosomal RNA and tRNA genes of the JH545 genome were masked before the analyses. CoverM was run with the following options: --mapper bwa-mem --methods relative_abundance --min-read-aligned-length 50 --min-read-percent-identity 95. Note that only the JH545 genome was used as the reference genome. The threshold value for "--min-read-percent-identity" was set to 95%, an ANI value widely used for species demarcation, as we wanted to estimate the relative abundance of metagenome reads that could be regarded as being from the same species as JH545. The list of metagenome samples is provided in Supplementary Data 2.

### Analysis of fatty acid composition

The analysis of cellular fatty acid methyl esters (FAMEs) was performed using the standard protocol provided by the MIDI/Hewlett-Packard Microbial Identification System. To extract FAMEs, cells of strain JH545 were harvested by centrifugation at $13,000 \times g$ for 1 h at the end of the exponential growth phase in a 400 mL volume of liquid culture (ASW5x media). The extracted FAMEs were saponified and methylated before being analyzed using a gas chromatograph (Agilent 7890 GC) with TSBA6 database from the Sherlock Microbial Identification System (MIDI) version 6.1.

### Proposal of ranks of the new taxa

**Description of "*Candidatus* Lucifugimonas" gen. nov.** *Lucifugimonas* (Lu.ci.fu.gi.mo'nas. L. fem. n. *lux, lucis*, light; L. fem. n. *fuga*, flight; L. fem. n. *monas*, a unit, monad; N.L. fem. n. *Lucifugimonas*, a monad that prefers dark habitats).

Aerobic, oligotrophic, and chemoheterotrophic. Gram-negative with a monoderm envelope. Cells are non-motile and dimorphic with short rods of ~0.8 × 0.4 μm and cocci of ~0.5 μm diameter. Do not form colonies on solid agar plates. Grows very slowly and reaches stationary phase at ~50 days of growth in artificial seawater medium. Light inhibits cellular growth. Major cellular fatty acids are summed feature 9 ($C_{17:1}$ ω9c and/or 10-methyl $C_{16:0}$), 10-methyl $C_{18:0}$, and $C_{16:0}$ (Supplementary Table 4). The genus "*Candidatus* Lucifugimonas" is assigned to SAR202 group Ia within the class *Dehalococcoidia* based on 16S rRNA gene phylogeny and whole genome phylogenomics. The type species of the genus is "*Candidatus* Lucifugimonas marina".

**Description of "*Candidatus* Lucifugimonas marina" sp. nov.** *Lucifugimonas marina* (ma.ri'na. L. fem. adj. *marina*, marine, of the sea).

In addition to the properties given in the genus description, the species is described as follows. Growth occurs at temperatures between 10 and 25 °C, but not at 4 °C or below, nor at 30 °C or above. Optimum growth temperature is 15–20 °C. Grows only in seawater-based liquid medium or artificial seawater medium. Cellular growth is enhanced by fucose, fuconate, fucono-1,4-lactone, rhamnose, rhamnono-1,4-lactone, rhamnonate, and ascorbate. The type strain, JH545, was isolated from epipelagic seawater off the coast of Garorim Bay at Tea-An, South Korea. The length of the complete whole genome sequence of the type strain is 3.08 Mbp with 51.8% of the DNA G + C content. GenBank accession number of the type strain is CP046146. Besides the type strain, whole genome sequences of strains JH639, JH702, and JH1073 belonging to this species are also available under GenBank accession numbers WMBD00000000, WMBE00000000, and CP046147, respectively.

**Description of "*Candidatus* Lucifugimonadaceae" fam. nov.** *Candidatus* Lucifugimonadaceae (Lu.ci.fu.gi.mo.na.da.ce'ae. N.L. fem. n. *Lucifugimonas*, a bacterial genus; *-aceae*, ending to denote a family; N.L. fem. pl. n. *Lucifugimonadaceae*, the *Lucifugimonas* family).

The description is the same as with the genus "*Candidatus* Lucifugimonas". The type genus is "*Candidatus* Lucifugimonas". Equivalent to GTDB f_UBA1328 (R207).

**Description of "*Candidatus* Lucifugimonadales" order. nov.** *Candidatus* Lucifugimonadales (Lu.ci.fu.gi.mo.na.da'les. N.L. fem. n. *Lucifugimonas*, a bacterial genus; *-ales*, ending to denote a family; N.L. fem. pl. n. *Lucifugimonadales*, the *Lucifugimonas* order).

The description is the same as with the genus "*Candidatus* Lucifugimonas". The type genus is "*Candidatus* Lucifugimonas". Equivalent to GTDB o_UBA1151 (R207).

### Reporting summary

Further information on research design is available in the Nature Portfolio Reporting Summary linked to this article.

## Data availability

All relevant data supporting the findings of this study are available within the paper and its supplementary information and data files. The 16S rRNA gene sequences of 24 isolates in SAR202 group I generated in this study have been deposited in the GenBank database under accession numbers OQ689977 to OQ690000. The whole genome sequences generated in this study are available in the GenBank database under accession numbers CP046146 (JH545), CP046147 (JH1073), WMBD00000000 (JH639), and WMBE00000000 (JH702). The genomic data are also available in the IMG/M database under genome IDs 2901382945 (JH545), 2917498938 (JH1073), 2892960865 (JH639), and 2892963810 (JH702). Source data are provided with this paper.

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

## Acknowledgements

This research was supported by High Seas Bioresources Program of Korea Institute of Marine Science & Technology Promotion (KIMST) funded by the Ministry of Oceans and Fisheries (KIMST-20210646 to J.-C.C.) and National Research Foundation of Korea (NRF) grants (NRF-2022R1A2C3008502, NRF-2018R1A5A1025077, and NRF-2021M3A9I4021431 to J.-C.C.; NRF-2022R1A6A3A01087360 to Y.L.) funded by the Ministry of Sciences and Information and Communications Technology, Korea.

## Author contributions

J.-H.S. and J.-C.C. planned and designed the initial isolation project. Y.L. and J.-H.S. performed the experiments. Y.L., J.-H.S., and I.K. analyzed the data. Y.L., I.K., S.J.G., and J.-C.C. wrote and revised the manuscript. I.K. and J.-C.C. supervised the project.

## Competing interests

The authors declare no competing interests.
