## [Peer Review File · Nature Communications]

Cultivation of Marine Bacteria of the SAR202 CladeReviewer #1 (Remarks to the Author):

This manuscript is very well written and represents an important step in our understanding of microbial diversity in the oceans. SAR202 is without a doubt one of the most important lineages on the planet and getting into culture and studying its growth will be of broad interest and thus I fully endorse this be published in Nature Comm. I have very few comments all of which are minor changes that should be made.

Ln 16 – I agree this is at the top of that list, however I think some might take offense to say they were at the “top”, maybe change to “among the top”.

Ln – 53 – not clear on “keen interest and technological advances” would take them off this list? Reword this sentence, make no sense to me.

Ln 435 – what version of GTDB-Tk was used?

Reviewer #2 (Remarks to the Author):

The manuscript by Lim et al. reports the isolation SAR202 bacterioplankton for the first time. This is a significant achievement in microbial ecology, given the ubiquity and abundance of this lineage in global oceans. I sincerely acknowledge their efforts and achievements. Using the isolated strains, the authors performed genome analysis and eco-physiological characterization. Although the manuscript is well-written and easy to follow the context overall, several points require logical/methodological clarification. The followings are my review comments.

Major comments:

1. Representativeness of the isolate among the SAR202 clade

The isolated strains all belong to group I of the SAR202 clade. However, group III dominates in the deep ocean, where SAR202 shows high relative abundance that initially attracted researchers. In this regard, the authors must be careful when generalizing the results based on the isolates to the whole SAR202 clade. Indeed, a previous study indicated a significant gap in the genetic repertoire between groups I and III (Saw et al. 2020). For instance, COG4948, which is overrepresented in group I genomes and one of the main subjects of this manuscript, is not that abundant and diverse in group III genomes. The existence of (helio-)rhodopsin is also not the case in group III and thus is unlikely the general feature of SAR202. The growth inhibition at low temperatures (Extended Fig. 3) is also unlikely for members inhabiting deep oceans. Linking the results with the molecular diversity hypothesis is intriguing. However, this should be carefully introduced because many people would imagine the processes in the deep ocean when it comes to this hypothesis and SAR202. Overall, given the broad phylogenetic range of the SAR202 clade, the characteristics of group I and that of the SAR202 clade should be clearly separated in the discussion.

2. Interpretation of the carbon compound-adding experiments (Fig. 5)

The authors' interpretation and aim of the experiment in Fig 5 are unclear to me. The result demonstrates the growth enhancement by the compounds but does not provide information on how these compounds are used by the cells. Unless showing there is no other predicted metabolic pathway to utilize these compounds, linking the results (Fig. 5a) with the overrepresentation of some COGs (Fig. 5b) is not logically convincing. Further, since the compounds are not the sole carbon source in the media, the results cannot conclude that these compounds are serving as carbon and energy sources (L. 237). Overall, the authors need to clarify the aim of the experiment and reasonably discuss what was demonstrated and what remains unknown in the experiment.

3. Method for read-mapping analysis

The authors estimated the relative abundance of JH545 across the ocean using metagenomic read mapping. Here, I could not evaluate the validity of the analysis due to the following reasons. First of all, RPKM is the value normalized by the number of mapped reads. This means that RPKM would change significantly depending on the size of the mapping reference. For instance, if other

dominant members (like SAR11) were included in the mapping reference, the RPKM would be estimated lower, as the number of mapped reads increases. With this regard, what was used for the mapping reference is unclear in the method section. Second and related, it is unclear whether the authors performed competitive mapping or not. If there is a close relative (ANI > ~90% (= the threshold of read mapping)) of JH545 in the sample, or if the JH545 have a conserved genomic region shared with many other bacteria, the reads from those non-JH545 genomes would be mis-mapped to JH545 and result in overestimation. (more information for competitive mapping is available here: https://instrain.readthedocs.io/en/master/important_concepts.html#handling-and-reducing-mis-mapping-reads). With this regard, the term "abundance of JH545" is inappropriate here because JH545 is a name of an isolated strain, and short-read mapping will never reveal the abundance of the same genotype. Seeing Fig. 1, I guess that the authors' read mapping analysis would also detect JH1073, JH639, JH702, even UBA7894, and other related species in the same genus/family. I would doubt the high RPKM at >1000m depth might be false positives derived from close relatives, not rhodopsin-holding members of JH545. Finally, the size fraction (filter pore size) of the mapped metagenomes is unclear. Differences in the size fraction may cause biases in the abundance estimation due to the different proportions of non-prokaryotic reads among the samples. Please clarify the method for the read-mapping analysis.

Minor comments:

L. 79: What was the subclade of the second abundant SAR202 in Extended Fig. 2? They are also abundant in the sample water but were not recovered as an isolate. Is there an explanation for this?

L. 107: Is "influence of horizontal gene transfer" important? Are there any important genes on the genomic islands?

L. 109 "shared similar features": How similar were they to each other? What was ANI between them? Is there a hypervariable region unshared between strains (this cannot be captured by metagenome and is one of the main advantages of using cultivated strain)? Do they share genomic synteny? I would suggest drawing a dot plot among the genomes if it shows something interesting.

L. 129-: The discussion of archaeellum is fascinating. Drawing a gene phylogenetic tree would further conclude the origin of the Chloroflexota archaeellum. Has this already been done by others? If not, I would suggest drawing the tree.

L. 283: Please cite the reference for the observation that Bacteroidia, Gammaproteobacteria, and Verrucomicrobiota are responsible for polysaccharide degradation.

L. 306-: Since the isolation is innovative, I would encourage the author to describe more about future perspectives in the conclusion section. What will become possible in the future by using this isolate? What remains unknown in the present study, and what should be done in the next step?

L. 377-: What was the initial cell density of this experiment?

L. 430 "ANIb": Is this typo?

Fig. 3: Please provide an explanation for the colors of the transporter illustrations.

Extended Fig 4: "as drawn by IMG" "integrated results" What do they mean? Please specify.

Extended Table 1: This table is hard to read as there are many horizontal lines. Please modify the format for better readability.

Authors' response to Reviewers' comments

Reviewer #1:

This manuscript is very well written and represents an important step in our understanding of microbial diversity in the oceans. SAR202 is without a doubt one of the most important lineages on the planet and getting into culture and studying its growth will be of broad interest and thus I fully endorse this be published in Nature Comm. I have very few comments all of which are minor changes that should be made.

Line 16: I agree this is at the top of that list, however I think some might take offense to say they were at the "top", maybe change to "among the top".

Response: We agree with your point, and we've changed "top" to "among the top" (line 16).

Line 53: not clear on "keen interest and technological advances" would take them off this list? Reword this sentence, make no sense to me.

Response: We revised the sentence that included this phrase to better convey what we are trying to say. We've wanted to emphasize that many prokaryotic groups are not cultured yet even though many researchers who recognize the importance of cultivation have applied novel techniques recently. The revised sentences are as follows (lines 54-57).

"Despite the recent surge in interest and technological advances in cultivation, a high proportion of prokaryotic groups remain uncultured. The SAR202 clade has no cultured isolates yet, making it one of the "most wanted" in culture and a key "target for cultivation".

Line 435: what version of GTDB-Tk was used?

Response: We used GTDB-Tk version 1.7.0. We added this version information in the manuscript. Please refer to line 465.

Reviewer #2:

The manuscript by Lim *et al.* reports the isolation SAR202 bacterioplankton for the first time. This is a significant achievement in microbial ecology, given the ubiquity and abundance of this lineage in global oceans. I sincerely acknowledge their efforts and achievements. Using the isolated strains, the authors performed genome analysis and eco-physiological characterization. Although the manuscript is well-written and easy to follow the context overall, several points require logical/methodological clarification. The followings are my review comments.

Major comments:

1. Representativeness of the isolate among the SAR202 clade

The isolated strains all belong to group I of the SAR202 clade. However, group III dominates in the deep ocean, where SAR202 shows high relative abundance that initially attracted researchers. In this regard, the authors must be careful when generalizing the results based on the isolates to the whole SAR202 clade. Indeed, a previous study indicated a significant gap in the genetic repertoire between groups I and III (Saw *et al.* 2020). For instance, COG4948, which is overrepresented in group I genomes and one of the main subjects of this manuscript, is not that abundant and diverse in group III genomes. The existence of (helio-)rhodopsin is also not the case in group III and thus is unlikely the general feature of SAR202. The growth inhibition at low temperatures (Extended Fig. 3) is also unlikely for members inhabiting deep oceans. Linking the results with the molecular diversity hypothesis is intriguing. However, this should be carefully introduced because many people would imagine the processes in the deep ocean when it comes to this hypothesis and SAR202. Overall, given the broad phylogenetic range of the SAR202 clade, the characteristics of group I and that of the SAR202 clade should be clearly separated in the discussion.

Response: We agree with the reviewer that i) many researchers have been originally interested in the SAR202 clade largely due to its abundance in the deep ocean, ii) diverse SAR202 groups show different depth distributions, with the SAR202 group III being the most abundant group in the deep ocean generally, and iii) there are evident genomic (and putatively metabolic) differences between the group III and group I to which our isolates belong.

With this in mind, we think we tried to indicate clearly in our original manuscript that our isolates belong to the group I. We note however that many readers of our manuscript are likely to regard the features we report as generally applicable to the whole SAR202 clade, especially because this is the first report about cultured SAR202 cells. Therefore, we revised our manuscript to make it clearer that the findings of this study are from the SAR202 group I isolates and group III members in the deep ocean can have different features.

First, we indicated in the first paragraph of the section on COG4948 that diverse SAR202 groups show group-specific pattern of paralog expansion as follows (lines 196-198).

“Many marine bacteria have minimal genomes with few paralogs, so the unusually large sets of paralogs present in diverse SAR202 groups, such as COG4948 and COG2141 in the groups I and III, respectively, attracted attention when they were first discovered.”

Further, we changed “The largest COG sets in the SAR202 genomes,” to “The largest COG sets in the SAR202 genomes of this study,” to indicate more clearly that the abundant COG sets we report applies to our isolates and group I (line 226).

Second, we added a short discussion in the section on the vertical distribution of the JH545 population, to indicate that we also need data from other SAR202 groups (such as group III) to better understand what underlies the prevalence of the SAR202 clade in the deep ocean as follows (lines 305-309).

“We note, however, that there are other SAR202 groups (e.g., group III) that are known to contribute substantially to the high relative abundance of the SAR202 clade in the deepest ocean regions. The isolation and characterization of a wider diversity of SAR202, especially lineages typical of the dark ocean, could propel future research that aims to reconstruct carbon chemistry and ecology in the dark ocean.”

Third, we also added a short discussion on the relevance of other SAR202 groups (such as group III) to molecular diversity hypothesis and carbon cycling/sequestration in the deep ocean just before the concluding remarks as follows (lines 324-328).

“It should be noted that carbon pools in the dark ocean could also be affected by other SAR202 groups. For example, SAR202 group III, which is known to be more abundant than group I in the dark ocean, has been suggested to contribute to the degradation and transformation of recalcitrant organic matter using an expanded repertoire of flavin-dependent monooxygenases.”

Fourth, we added a sentence in the section on HeR to indicate that the existence of HeR may indicate the adaptation of some SAR202 members (such as our isolates) to surface ocean, rather than general features of the whole SAR202 clade as follows (lines 176-178).

“Given that no HeR has been found in SAR202 group III, which is abundant in the dark ocean, the possession of HeR by JH545 isolated from coastal surface water may indicate adaptation to the euphotic habitat.”

Fifth, we added a short discussion in the SAR202 growth section to describe the implication of growth characteristics of JH545 observed at 4 °C as follows (lines 89-92).

“Very little growth followed by gradual decline at 4 °C (Supplementary Fig. 3) suggests that strain JH545 belonging to the SAR202 group I may not be well-adapted to deep sea, where other members of the SAR202 clade (e.g., group III) are prevalent.”

Finally, we'd like to note that we believe studies of surface-adapted lineages of marine bacterial clades that have long been regarded to be largely confined to the dark ocean (e.g., our SAR202 isolates) would be interesting and essential to understanding these clades. Recently, several genome-centric metagenomic studies were performed on such clades, including the SAR324 (Boeuf *et al.*, 2021, *Microbiome*), SAR406 (Getz *et al.*, 2018, *MBio*), and SAR202 (Saw *et al.*, 2020, *MBio*) clades. These clades represent phylum-level groups (SAR324; p__SAR304, SAR406; p__Marinisomatota) or a superorder (SAR202; in this study) in GTDB. Reflecting the wide phylogenetic diversity of these clades, these studies showed that metabolic potentials are highly diversified among the members of each clade and revealed that some members are found preferentially in euphotic zones, as opposed to early generalizations that were based on broad phylogenetic categorizations. Therefore, more thorough understanding on these important clades will require the investigation of diverse members that show differential depth preferences.

2. Interpretation of the carbon compound-adding experiments (Fig. 5)

The authors' interpretation and aim of the experiment in Fig 5 are unclear to me. The result demonstrates the growth enhancement by the compounds but does not provide information on how these compounds are used by the cells. Unless showing there is no other predicted metabolic pathway to utilize these compounds, linking the results (Fig. 5a) with the overrepresentation of some COGs (Fig. 5b) is not logically convincing. Further, since the compounds are not the sole carbon source in the media, the results cannot conclude that these compounds are serving as carbon and energy sources (L. 237). Overall, the authors need to clarify the aim of the experiment and reasonably discuss what was demonstrated and what remains unknown in the experiment.

Response:

The reviewer raises an important point that we have addressed by altering the manuscript. We want readers to understand that research with cells that are challenging to cultivate frequently requires using different approaches. The data we report demonstrate a pronounced growth response to organic carbon substrates predicted by genome reconstruction, but do not enable us to determine if they were used as carbon or energy sources or benefited the cells by some other mechanism. Therefore, we added further explanations to the revised manuscript as follows (lines 249-262).

" We tested the growth response of JH545 cells to external metabolites that were predicted to be substrates for the catabolic pathways proposed above. In these experiments we used a defined medium based on artificial seawater, to which we added the same cofactors and organic compounds that were used for the original isolation of the strains. We did not examine whether the tested substrates could serve as sole carbon sources because of the very low growth rate of the cells and because cells adapted to oligotrophic ecosystems often exhibit reduced metabolic flexibility in comparison to copiotrophic cells when challenged with simplified carbon mixtures. All substrates tested, including L-fucose, L-rhamnose, their lactone and acid forms, and ascorbate enhanced the growth of strain JH545 in artificial seawater media (Fig. 5a). With increases in growth rates, cell densities in late exponential phase (~35 days of incubation) were more than 10 times higher in substrate-amended cultures, although the cultures reached similar densities in stationary phase. Ascorbate was tested because an ascorbate degradation pathway requiring a COG4948 enzyme was nearly complete in the JH545 genome (Supplementary Fig. 9)."

For context, the sole carbon source concept has not transferred well to experimental work with numerically important oligotrophs. We can draw parallels to one of the most well-developed model systems for oligotrophic cell culture, SAR11, which has no sole carbon sources, and never achieves high cell densities or rapid growth rates but has been proven to metabolize a wide range of compounds. Substantial investment in developing these cells as a model system has uncovered properties that may partly explain these behaviors. Reduced metabolic connectivity and regulation have been proposed to be adaptive for cells replicating in a dilute and complex organic compound milieu (e.g. Noell *et al.*, 2023, *Microbiol. Mol. Biol. Rev.*).

We hope the reviewer will agree that the cellular responses in growth experiments to exometabolite compounds predicted by the metabolic reconstruction are evidence in support of the conceptual model we propose to explain the presence of paralog families in these cells.

To clarify the arguments for this interpretation, we further revised the manuscript as follows (lines 240-244):

“Although both rhamnose and fucose are known to be degraded via pathways involving phosphorylation in many organisms, these kinase-dependent pathways were not found in the genomes. This suggests that the reconstructed non-phosphorylative pathways shown in Fig. 5b might serve as the only catabolic routes for both sugars in these strains.”

3. Method for read-mapping analysis

The authors estimated the relative abundance of JH545 across the ocean using metagenomic read mapping. Here, I could not evaluate the validity of the analysis due to the following reasons. First of all, RPKM is the value normalized by the number of mapped reads. This means that RPKM would change significantly depending on the size of the mapping reference. For instance, if other dominant members (like SAR11) were included in the mapping reference, the RPKM would be estimated lower, as the number of mapped reads increases. With this regard, what was used for the mapping reference is unclear in the method section. Second and related, it is unclear whether the authors performed competitive mapping or not. If there is a close relative (ANI>~90% (= the threshold of read mapping)) of JH545 in the sample, or if the JH545 have a conserved genomic region shared with many other bacteria, the reads from those non-JH545 genomes would be mis-mapped to JH545 and result in overestimation. (more information for competitive mapping is available here: https://instrain.readthedocs.io/en/master/important_concepts.html#handling-and-reducing-mis-mapping-reads). With this regard, the term “abundance of JH545” is inappropriate here because JH545 is a name of an isolated strain, and short-read mapping will never reveal the abundance of the same genotype. Seeing Fig. 1, I guess that the authors’ read mapping analysis would also detect JH1073, JH639, JH702, even UBA7894, and other related species in the same genus/family. I would doubt the high RPKM at >1000m depth might be false positives derived from close relatives, not rhodopsin-holding members of JH545. Finally, the size fraction (filter pore size) of the mapped metagenomes is unclear. Differences in the size fraction may cause biases in the abundance estimation due to the different proportions of non-prokaryotic reads among the samples. Please clarify the method for the read-mapping analysis.

Response: We understand the reviewer's concerns with the many factors that influence read-mapping metrics. We repeated the analysis with CoverM, a widely used tool to estimate the abundance of genomes in metagenome samples, and replaced the original figures with new figures that use units of "relative abundance

(%)" instead of "RPKM". The results are generally similar and support the conclusion that the "SAR202 isolates represent a cell type that is common in the euphotic zone but relatively more abundant in the dark ocean".

In the original manuscript, BlastN was used for read-mapping. Due to the slow speed of Blast and limitations in computing resources available to us, we subsampled 1 million reads from each metagenome before read mapping. We decided subsampling was not suitable for this study, since randomness in subsampling might have unexpectedly large effects on the results for the genomes whose abundances are very low. Many SAR202 genomes have already been shown to be rare in a previous study (please refer to Saw *et al.*, 2020, *MBio*, especially Fig. 6 and its shading scale). Therefore, we determined to use the entire metagenome data without subsampling for reanalysis, which required more scalable tool such as CoverM. The procedures we followed for fragment recruitment are presented in the revised manuscript (lines 519-530).

We note that the revised results obtained using CoverM show patterns generally similar to those we presented in the original manuscript, but more scatter in the results from marine trenches (Fig. 6b). Therefore, we changed the "geom_smooth" options for drawing regression (smoothing) lines in the plot, from "lm" to "gam" (Fig. 6b). To reflect this change, we also revised sentences describing the results of marine trenches as follows (lines 271-273) and the legend for Fig. 6.

" In metagenomes from several marine trenches, the relative abundance of the JH545 population generally increased with increasing depth above the hadopelagic zone (below 6,000 m), where their relative abundance declined (Fig. 6a–b)"

We also revised Fig. 6c (for Tara Oceans) so that bubble colors correspond to the oceans. Although this was not requested by the reviewers, we revised the plot because we believe that this change will help readers assess the global distribution of JH545 population more easily. The legend for Fig. 6c was revised accordingly.

We acknowledge that we did not explain clearly about how we calculated the RPKM values in our original manuscript, which was different from the conventional calculation. As pointed out correctly by the reviewer, usually, RPKM is normalized by the total number of reads that are mapped to any one of reference genomes included in analyses, so RPKM of a specific genome is affected by how many other genomes were used together and also by the abundance of other genomes. But, the RPKM in our original manuscript was not normalized by the total number of reads that are mapped, but was normalized by the total number of reads that are used for mapping. Because we used 1 million subsample reads for each metagenome in the original manuscript, the number of reads mapped to the JH545 genome was divided by 1 million reads for all metagenomes. Therefore, we think that the RPKM values in our original manuscript were free from the issues raised by the reviewer. We acknowledge that our way of calculating RPKM may be confusing to many readers because our calculation differed from widely used approaches. Therefore, we defined our RPKM in the original manuscript as "the number of recruited reads per kilobase of genome per million total reads" (lines 250-251 of the original manuscript). But, this phrase was not enough to indicate clearly the difference between our use of RPKM and more widely used RPKM. Because we used CoverM with the "--methods relative_abundance" option in the revised manuscript, these issues are no longer relevant. Note that CoverM has also "rpkM"

options, but we did not use this option.

We used only the JH545 genome for read mapping analysis both in the original and revised manuscript, thus we used non-competitive mapping. While we used a cutoff value of 90% sequence identity in the original manuscript, we used 95% as "--min-read-percent-identity" option of CoverM in the revised manuscript as follows (lines 526-529).

"The threshold value for "--min-read-percent-identity" was set to 95%, an ANI value widely used for species demarcation, as we wanted to estimate the relative abundance of metagenome reads that could be regarded as being from the same species as JH545."

We took this approach (using only the JH545 genome with a cutoff value of 95%) because we think it serves our purpose well, i.e., estimating the relative abundance of the species-level population represented by JH545. We agree with the reviewer that the term "abundance of JH545" was used inappropriately in our manuscript. In the revised manuscript, we tried to use the term "JH545 population" consistently instead of "JH545", after indicating that we will use the term to describe species-level population represented by JH545 in the first sentence of the section titled "SAR202 isolates represent a cell type that is common in the euphotic zone but relatively more abundant in the dark ocean". Please note that we revised several phrases in this section to convey our message more clearly (lines 266-308).

Because we used 95% as the cutoff value of read mapping and because the four SAR202 genomes of this study have >99.9% of ANI values among themselves (refer to our response to another comment below), we agree with the reviewer that we detected all three strains (i.e., JH1073, JH639, JH702) and closely related lineages. But we think that this would not invoke any problems in the interpretation of our results, because these four strains clearly belong to the same species (based on ANI) and we wanted to estimate the abundance of species-level population represented by JH545. We think that the metagenome reads similar to the UBA7894 genome will not have a large impact because the ANI value between JH545 and UBA7894 is ~75%, far below the cutoff of 95% (by JSpeciesWS). Because we used 95% as a cutoff value, we think that it is likely that the metagenome reads that are recruited from >1000m depth are from SAR202 cells that putatively belong to the same species as JH545.

Following the suggestion, we've added the size fraction of the metagenomes in Supplementary Table 4 of the revised manuscript. The metagenomes of marine trenches (Fig. 6a-b) were from >0.22- μ m. The metagenomes of Tara Oceans and station ALOHA were largely from 0.22–3.0 μ m (70 samples), except for 12 samples from the two Tara Oceans stations and station ALOHA (0.22–1.6 μ m). We agree with the reviewer that the size fraction may have effects on the estimation of relative abundance. But we think that the size fraction would have a very little, if any, effects on the findings of this study. First, the metagenomes of trenches were not compared to the metagenomes of Tara Oceans and station ALOHA. We analyzed only the depth profiles within trench metagenomes that are from the same size fraction. Second, regarding the metagenomes of Tara Oceans and station ALOHA, i) most of stations and samples (25 stations among 28; 70 samples among 82) were

from the same size fraction (0.22–3.0 μm), ii) samples from the same stations were from the same size fraction, and our analyses focused on the depth profiles rather than comparison between the stations, iii) both size fractions (0.22–3.0 μm and 0.22–1.6 μm) are the fractions widely used for analyses of prokaryote.

Minor comments:

Line 79: What was the subclade of the second abundant SAR202 in Extended Fig. 2? They are also abundant in the sample water but were not recovered as an isolate. Is there an explanation for this?

Response: Before responding to this comment, please note that the previous ‘Extended Fig. 2’ has been changed to ‘Supplementary Fig. 2’.

We performed BlastN using the ASV sequences as queries against the SSU rRNA gene sequences of GTDB (R202). Assignment of putative GTDB taxonomy to the ASVs was performed based on the GTDB taxonomy of the best hits. All the four ASVs of the SAR202 clade, including the second abundant one, were found to belong to the SAR202 group I (o__UBA1151). While the most abundant ASV corresponding to the isolates is affiliated with f__Bin127 (group Ia; refer to Fig. 1), the second abundant ASV is affiliated with f__TMED-70 (group Ib or Ic). This classification results have been added to Supplementary Fig. 2b of the revised manuscript, with a description of analysis methods in the legend. The SSU rRNA gene sequences we used are available at https://data.gtdb.ecogenomic.org/releases/release202/202.0/genomic_files_all/ssu_all_r202.tar.gz.

Unfortunately, we don’t have any explanation on why we couldn’t culture an isolate(s) corresponding to the second abundant ASV. We just suspect that differences in viability (Henson *et al.*, 2020, *Appl. Environ. Microbiol.*) and metabolic activity (Munson-McGee *et al.*, 2022, *Nature*) between the SAR202 populations might have affected the cultivation results.

Line 107: Is “influence of horizontal gene transfer” important? Are there any important genes on the genomic islands?

Response: We deleted the phrase "horizontal gene transfer" from the revised manuscript.

Although horizontal gene transfer (HGT) is important in prokaryotic evolution in general, it is not necessary here. We note that most of the genes on the genomic islands predicted by IslandViewer 4 seem to encode hypothetical proteins. Genes typical of mobile genetic elements, such as recombinase and transposase, were found, suggesting HGT is an evolutionary force that impacts these genomes. But, since no genes of metabolic or ecological relevance to our narrative were not found, this topic can be left for another day.

Line 109 “shared similar features”: How similar were they to each other? What was ANI between them? Is there a hypervariable region unshared between strains (this cannot be captured by metagenome and is one of the main advantages of using cultivated strain)? Do they share genomic synteny? I would suggest drawing a dot plot among the genomes if it shows something interesting.

Response: Our intent in the reference to “shared similar features” was to convey that the four genomes are

nearly identical to each other in terms of genome-inferred metabolic features (potentials). We found little difference between the genomes in reconstructed metabolic pathways. Also, the functional profiles obtained from the IMG-ER annotation (i.e., number of genes assigned to the entries of PFAM, COG, KO, and TIGRFAM) were nearly identical among the genomes. We admit that the phrase did not convey well what we wanted to describe. Therefore, we revised the sentence as follows (lines 114-116).

“Genome-inferred metabolic features were nearly identical among the four genomes (e.g., COG profiles; Supplementary Table 3), leading us to focus on one of them, JH545, for further analysis (Fig. 3)”

The ANI values among the genomes were >99.9%, which was indicated in line 101 of the original manuscript (line 107 in the revised manuscript). The genomes shared genomic synteny across the whole genome lengths. A MUMmer (nucmer) plot of the two complete genomes (JH545 and JH1073) shows well this synteny (refer to the figure below). Several short regions showing discontinuity in the plot coincided largely with the predicted genomic islands, indicating a very little differences between the genomes are related to mobile genetic elements (MGEs). We think that MGEs in the SAR202 genomes is an interesting subject for future investigation. But, we decided analysis of the MGEs was outside of the scope of this study, since no genes of apparent metabolic or ecological relevance were found in the MGEs.

Line 129: The discussion of archaellum is fascinating. Drawing a gene phylogenetic tree would further conclude the origin of the *Chloroflexota* archaellum. Has this already been done by others? If not, I would suggest drawing the tree.

Response: Thank you for this insightful comment. As far as we know, no one has performed phylogenetic analysis of archaella genes including those from bacterial genomes. Following the suggestion, we have constructed a maximum likelihood phylogenetic tree of FlaB (archaeal flagellin), which was included in the

revised manuscript as Supplementary Fig. 7. This tree confirmed our inference that the *Chloroflexota* archaea were originated from *Archaea*. Accordingly, we added a short explanation on the tree to the revised manuscript as follows (lines 155-158).

“A phylogenetic analysis of FlaB sequences found in strain JH545, several *Chloroflexota* genomes, and representative archaeal isolates, confirmed this inference. The FlaB sequences of *Chloroflexota* formed a monophyletic clade, which was located as a sister clade of an archaeal FlaB group (Supplementary Fig. 7)”

We’ve also added the analysis procedures we used in the section “Analyses of the HeR and FlaB” in “Methods” (lines 476-487).

Line 283: Please cite the reference for the observation that *Bacteroidia*, *Gammaproteobacteria*, and *Verrucomicrobiota* are responsible for polysaccharide degradation.

Response: We cited the following review paper, with the statement of “references therein” (line 298).

Arnosti, C. *et al.* The biogeochemistry of marine polysaccharides: sources, inventories, and bacterial drivers of the carbohydrate cycle. *Annu. Rev. Mar. Sci.* **13**, 81-108 (2021).

Please note that the number of references is already well over the limit of *Nature Communication* (70). Therefore, we determined to cite only a recent review paper in which readers can find many original research articles relevant to this issue.

Line 306: Since the isolation is innovative, I would encourage the author to describe more about future perspectives in the conclusion section. What will become possible in the future by using this isolate? What remains unknown in the present study, and what should be done in the next step?

Response: Following the suggestion, we added a description on what we think would be interesting research questions that can and should be addressed in future studies as follows (lines 340-348).

“Cell cultures of novel prokaryotes provide opportunities to study a wide range of cellular properties that cannot be determined from genomes alone. Interactions of these slowly growing cells with organic carbon exometabolites, and their unexplained sensitivity to light, are promising avenues for future work. Studies of SAR202 archaea function and integration with the cell envelope of SAR202 may provide clues into the ecology and cell architecture of these unusual cells. The findings we report provided surprising insights into challenges that long-stalled SAR202 cultivation. Whether the diversity of not-yet-cultured lineages of SAR202 cells inhabiting the dark ocean can be explained by similar properties remains to be seen”

Line 377: What was the initial cell density of this experiment?

Response: The initial cell densities of this temperature test ranged from 1.11×10^4 to 1.51×10^4 cells mL⁻¹

(Supplementary Fig. 3 of the revised manuscript).

Line 430: “ANlb”: Is this typo?

Response: “ANlb” is not a typo. We used JSpeciesWS platform (Richter *et al.*, 2015, *Bioinformatics*; <https://jspecies.ribohost.com/jspeciesws/#home>) to calculate ANI values. JSpeciesWS provides the two options for ANI calculation: ANlb and ANlm. While ANlb uses BLAST as its core alignment program, ANlm uses MUMmer (Richter and Rosselló-Móra, 2009, *Proc. Natl. Acad. Sci. U.S.A*). We indicated “ANlb” in our manuscript to make it clear that we used ANlb (not ANlm) in JSpeciesWS platform.

Fig. 3: Please provide an explanation for the colors of the transporter illustrations.

Response: We are sorry that we did not provide information on the colors of transporters. In the original manuscript, the colors of the transporters corresponded to the functional classification as follows: pink, ABC transporters; yellow, voltage-gated ion channels; cyan, ion channels (uniporters); light green, ion channels (antiporters or symporters); light blue, active transporters.

During the revision, however, we came to think that this classification might be too detailed. Therefore, we determined to classify the transporters into the two categories, i.e., ABC transporters and others. In the revised Fig. 3, ABC transporters and others are colored in pink and blue, respectively. We also added this color designation in the figure legend (lines 820-821). In addition, we’ve added arrows indicating the direction of transport to ABC transporters, which were also missing in the original manuscript.

Extended Fig. 4: “as drawn by IMG” “integrated results” What do they mean? Please specify.

Response: Please note that the original Extended Fig. 4 has been changed to Supplementary Fig. 5 in the revised manuscript. What we wanted to indicate by using the phrase “as drawn by IMG” is that this genome map has not been drawn by us but was produced by IMG-ER annotation of the genome. We agree with the reviewer that the phrase was not clear. Therefore, we deleted the phrase. Instead, we added a separate sentence to explain this point at the last part of the legend of Supplementary Fig. 5a as follows (lines 199-200).

“This genome map was produced by IMG-ER annotation.”

What we wanted to indicate by using the phrase “integrated results” is that the genomic islands (GI) presented in Supplementary Fig. 5b is the aggregation (union) of the islands predicted by several prediction methods implemented in IslandViewer 4. IslandViewer 4 uses several methods to predict GIs. The main output of IslandViewer 4 shows the aggregation (union) of GIs predicted by all methods, in addition to GIs predicted by each method, with different colors. In both the original and revised manuscripts, we presented only the aggregation (union) of GIs. We agree with the reviewer that the phrase “integrated results” was not clear. Therefore, we rephrased the legend as follows (lines 202-203).

“The outermost ring shows the positions of genomic islands predicted by several methods implemented

in IslandViewer 4.”

Extended Table 1: This table is hard to read as there are many horizontal lines. Please modify the format for better readability.

Response: Following your suggestion, we modified the format of this table. The modified table is presented as Supplementary Table 1 in the revised manuscript. We tried our best to remove as many horizontal lines as possible. We hope that readers would have no or little difficulty in reading this table.

--- END ---

Reviewer #2 (Remarks to the Author):

I acknowledge the authors' sincere response and work to revise the manuscript. I do not have additional comments. I believe this paper will be one of the milestones in aquatic microbial ecology.